# Lessons Learned on Geosynthetics Applications in Road Structures in Silesia Mining Region in Poland

**Jacek Kawalec** [1] , **Marcin Grygierek** [1] , **Eugeniusz Koda** [2] **and Piotr Osiński** [2,*]

[1] Department of Geotechnics and Roads, Faculty of Civil Engineering, Silesian University of Technology, 44-100 Gliwice, Poland; jacek.kawalec@polsl.pl (J.K.); marcin.grygierek@polsl.pl (M.G.)

[2] Department of Geotechnical Engineering, Faculty of Civil and Environmental Engineering, Warsaw University of Life Sciences, 02-776 Warsaw, Poland; eugeniusz_koda@sggw.pl

* Correspondence: piotr_osinski@interia.pl; Tel.: +48-22-535-213

**Abstract:** The paper aims at the research of pioneering applications of geosynthetic materials used for improvement of anthropogenic material for road contraction challenges in Poland. The presented case study concerns a road embankment construction process within an area of underground mining coal extraction for which significant deformations have been frequently recorded. To improve the bearing capacity of the structure base, the geosynthetic materials were used. The question; however, was how the anthropogenic materials, filling the embankment, will interact with each other over time. The assessment of the structural condition of the motorway surface was performed using the falling weight deflectometer and the calculated modules based on the back-analysis method. They confirmed the effectiveness of the geosynthetics used in the study. They also revealed that the mining exploitation, with simultaneous use of aggregate stabilization with geogrids, did not cause significant changes in the stiffness of the pavement layers. All observations, based on both field and laboratory tests, did not show any negative impact of anthropogenic soils on the structural behavior of geogrids.

**Keywords:** geosynthetics; anthropogenic materials; stabilisation by geogrid

## 1. Introduction

In Europe in late 80's, after the Iron Curtain collapsed, such countries as Poland started a new chapter in economical development. The infrastructure was one of the key areas where Poland was far behind Western Europe standards. Almost all the construction technologies were outdated, non-efficient, and in many cases unsafe for use [1–3]. Becoming open to new technologies and structural materials provided a strong impact for modernization of the entire construction industry, and until nowadays this trend is still being observed [4–8]. Lack of planning and poor economy management resulted in a huge production of waste, both municipal and industrial, which in many cases were deposited without any control [3–5]. The waste deposition became a serious issue, mainly in well-developed regions like Warsaw or Silesia, where the amount of by-products was rapidly increasing. A typical solution of the problem, back in the days was just enlarging the area used to deposit waste materials. In those days, the use of materials such as geosynthetics were barely noticeable [9–13]. Until 1990's only a few non-woven textiles manufacturers were operating, but the material quality and durability was insufficient for large scale road construction projects. Simultaneously, in Poland a boom in new geosynthetics applications has started. Now, after over two decades of gaining experience on a number of infrastructure projects, we are able to make scientifically based conclusions and comments on most of them.

The present paper describes a case study on road construction works on post-mining areas. It concerns an underground mining coal extraction site where a new motorway was planned to be constructed. However, due to significant deformation of the ground surface, caused by former mining excavations, the sophisticated ground improvement methods were necessary to be applied. For the linear structures, like roads, or other geotechnical investments the mining activity is a major safety threat, thus the need of maintenance efforts increase significantly [14–17].

Recently, a peculiar revolution in the field of understanding and describing the geogrid interaction with the unbound aggregate has been observed. It mainly concerns the road construction and ground mechanical condition improvement. An improved performance of the pavement due to the geosynthetic reinforcement and bearing capacity improvement [18–23] depends on main mechanisms [24–26]:

- Lateral confinement of the unbound aggregate in the base course;
- Increased bearing capacity;
- Tensioned membrane effect.

The road pavement is made of layers, which distributes traffic generated loads, disabling excessive deformation of the subsoil. The major role of a road pavement is bearing high frequency loads. The pavement works provide a very small range of deformation, between $10^{-6}$ and $10^{-3}$ m, when considering the vehicles load and stiffness of pavement layers [3,11]. The time gap between a road opening and the asphalt resurfacing is determined by the fatigue life. A depletion of the fatigue life of the pavement causes characteristic cracks and structural rutting. A significant issue when constructing a road is the location within mining areas where subsoil deformation occurs, including deformation of the surface. Mining deformations usually affect working conditions of the pavement layers, cause pavement stiffness reduction, and the decrease of fatigue life. The deformations of the ground surface can take the form of continuous [27] or discontinuous mechanisms [28–31]. The cracks observed on the road pavement as a result of the subsoil deformations are presented in Figure 1.

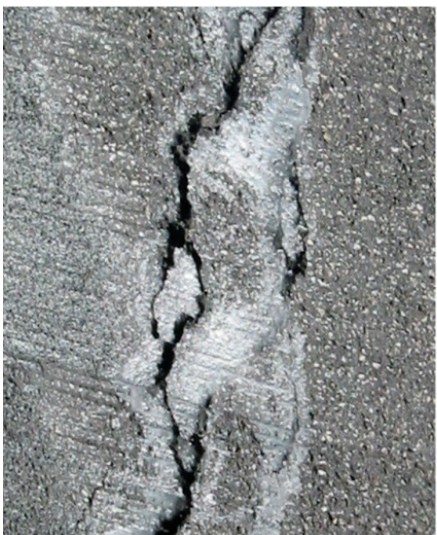 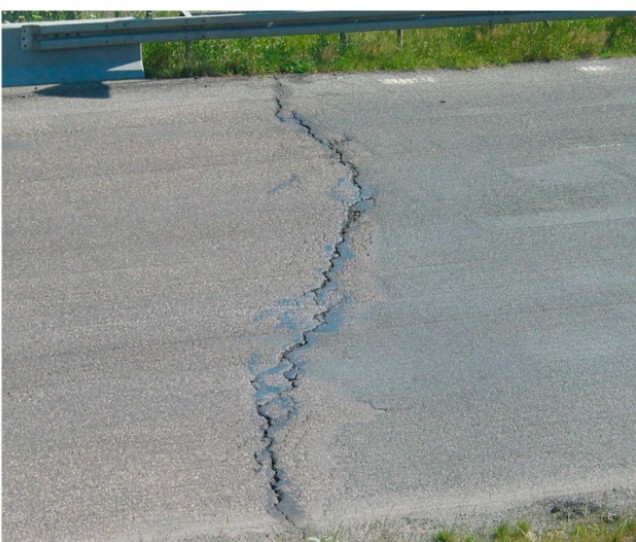

**Figure 1.** Typical damages observed on pavements.

In the case of continuous deformation, there is a subsidence on the surface of the ground on which the continuity of the subsurface rock layer does not cease.

Such subsidence, in general, is described by several surface deformation indicators among which the horizontal deformation ε [mm/m] should be distinguished, as they are included in the design of pavements, particularly in mining areas [27]. The horizontal deformation determines changes in the stiffness of the subsoil and pavement layers. On the convex part of subsidence, the loosening effect

(stretching) is activated, and in the bottom part of the subsidence, the re-compaction (compression) process may occur. The presentation of specific indicators usually affecting the subsidence is given in Figure 2. These are the conditions that always need to be considered when analyzing the working conditions of the road surfaces designed in post-mining areas.

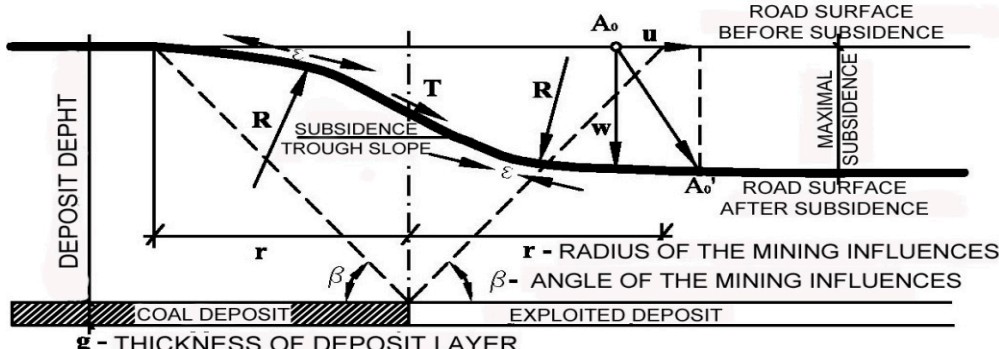

**Figure 2.** Mining induced subsidence trough: w—vertical displacement (m), T—inclination (mm/m), R—radius of curvature (km), $\varepsilon$—horizontal strain (mm/m), u—horizontal displacement (m) [32].

The impact of mining deformation can affect the pavement by reducing the modules and reducing the fatigue life of the pavement. For non-reinforced pavement (i.e., no geogrid), loosening deformations cause a significant increase of the horizontal deformations due to significant reduction of the layers' rigidity and the subsequent fatigue life. The deformation ranges, experienced in the past and analyzed by authors on a number of different case studies [32–34], are presented in Figure 3. The case studies represented different pavement categories (from 1 to 7) and concerned two types of construction layer compositions, for mining and non-mining areas [34].

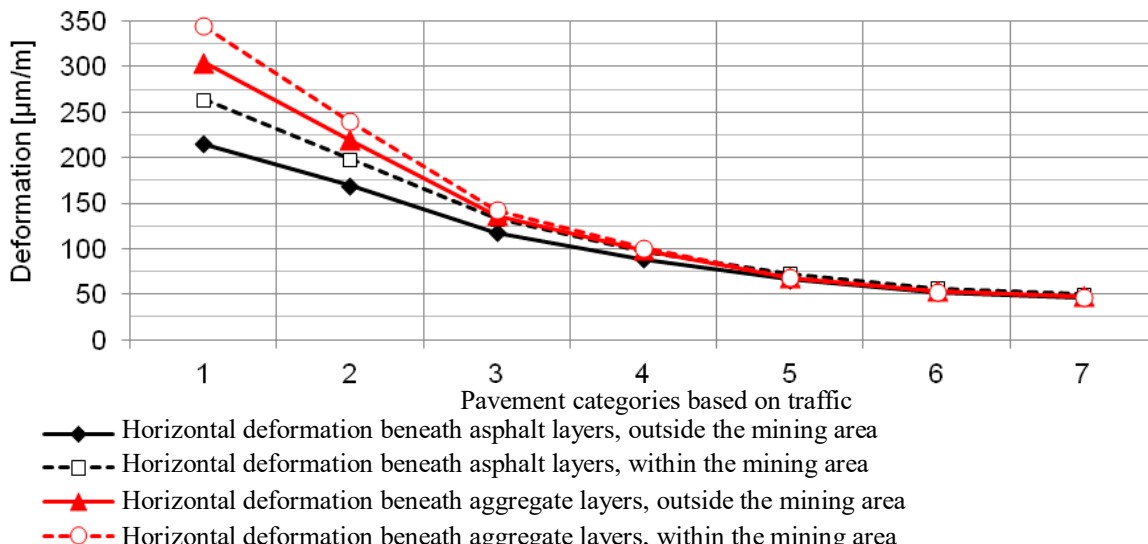

**Figure 3.** Horizontal deformation in selected pavement layers taking into account mining impact.

## 2. Materials and Methods

### 2.1. Stabilisation of the Embankment Base

In this particular application, a base layer of the embankment was placed over the geomattress, made of unbound aggregates stabilized with two layers of a monolithic geogrid. Due to the project requirements the aggregates of anthropogenic waste material was made of polypropylene [34].

Such a geomattress creates a quasi-stiff platform at the base, reducing most deep mining deformations on a surface, and allowing rapid construction. The installation of the geomattress is shown in Figure 4.

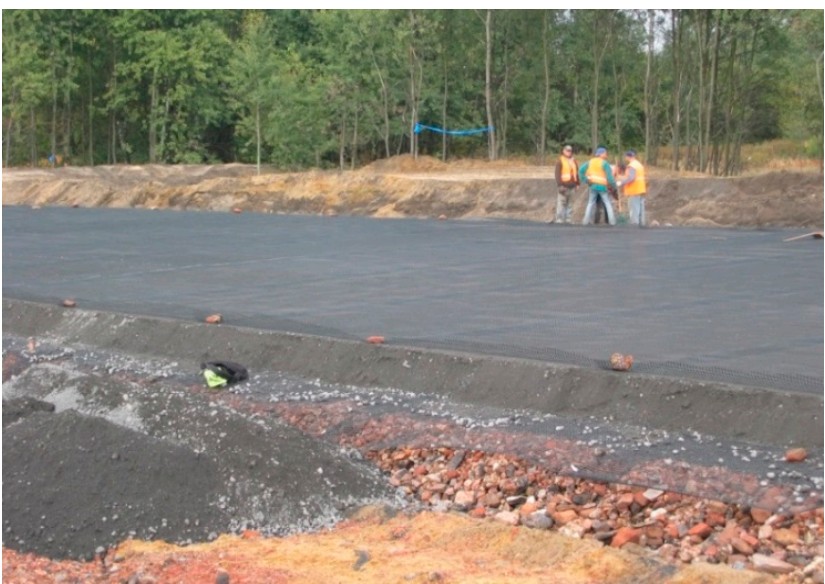

**Figure 4.** Installation of the geomattress, made of two layers of monolithic geogrids and anthropogenic aggregates.

The fill material for the geomattress in most cases was either slag or burned mining shale. Below the slag, the red colored aggregate was burnt mining shale. The stabilizing effect of the geomattress was achieved by interlocking the aggregate particles in apertures of the geogrid. It was achieved only if the geogrid was stiff and the aperture shape was not deformable under load, which was applied during the compaction of the aggregate layer. The interlocking effect of the geogrid makes the aggregate layer confined for some depth [10,27]. In case of mining influences, it becomes a barrier for a deformation propagation in directions from the below to the above layers.

*2.2. Evaluation of Mining Waste Mechanical Parameters*

Part of the complex preparation of the motorway design was the determination of the strength parameters of the mining waste materials that were to be used for the construction of an embankment body. In this case a full scale in-situ test, based on the method of trial loading was performed [29–32]. The applied methodology guaranteed a comprehensive distribution of the stress paths that were crucial for the proper identification of the design parameters. With reference to the trial loading of slopes, the in-situ tests enabled appropriate simulation of the failure process, incomparable to any laboratory tests or numerical analyses. The technical issues here referred to difficulties in achieving boundary equilibrium, corresponding with the most realistic estimations of the strength parameters: the internal friction angle ($\varphi$) and the cohesion ($c$) [33]. The tests performed on an experimental embankment allowed the measurement of the "load-settlement" relationship. It was necessary for further analyses by means of an elastic—ideally the plastic soil model.

The model was used to measure the slope geometry before and after the failure, taking into account a destructive load. The loading of the top surface of the slope with a pile of pavement slabs with a measurement of the settlements after each incremental step was considered to be the optimum solution. To be capable of considering the problem as linear (planar strain), a load was assumed to be applied on the length of a minimum of three pavement slabs, used as a surcharge. The model is presented in Figure 5.

The computations included estimations of the $\varphi$ and $c$ values, as parameters of an elastic (ideally plastic) Coulomb–Mohr model, based on back analyses applying the finite element method (FEM)

method and the Janbu method, respectively. The results obtained for the mechanical parameters, using the modeling, and also the physical parameters are listed in Table 1.

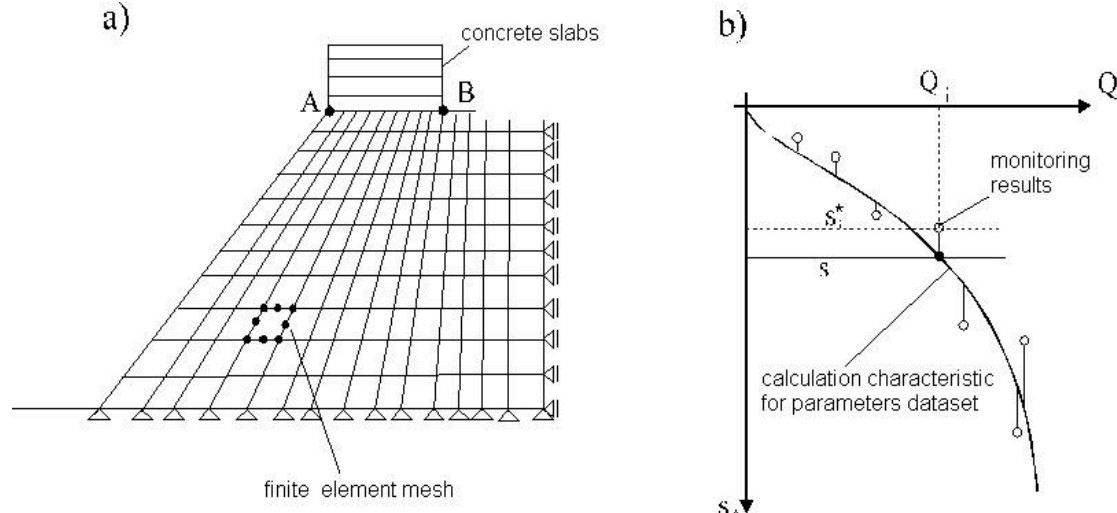

**Figure 5.** Finite element method (FEM) model used for mechanical parameters determination: (**a**) Model geometry; and (**b**) calculation characteristic curve of mechanical parameters referring to monitoring results.

**Table 1.** Physical and mechanical parameters (based on back analysis) of the filling material.

| Characteristics | Embankment Material | Aggregate in Mattresses |
|---|:---:|:---:|
| *physical* | | |
| dmax (mm) | <150 | 63 |
| $U = d_{60}/d_{10}$ (-) | 8.6–41.0 | 11.4–15.3 |
| Type | Unburned coal mining shales | Blast furnace slag |
| Carbon cont. $f_C$ (%) | 8.8 | n/a |
| $w_{opt}$ (%) ac. Proctor | 9.65 | 25.7 |
| $\rho ds$ (g/cm$^3$) ac. Proctor | 1.85 | 1.46 |
| Passive capillary (m) | 1.75 | n/a |
| Frost resistance (%) | 71 | 99.7 |
| *mechanical* | | |
| Internal friction angle $\varphi$ [°] | 22.7 | 40 |
| Cohesion c [kPa] | 50 | 4 |

The modeling part refers directly to the reliability assessment of waste strength parameters results, obtained on the basis of trial loading of the road embankment slope in the natural scale. The full-scale trial loading test was carried out using 18,000 tons of material on the 100 m long, 20 m wide, and 5 m high road embankment, presented in Figure 6.

The results of the strength parameters evaluation, with relatively low values of standard deviation, were used to design a motorway embankment in a section located near the mine. In 1990, the Polish government decided to construct a new section of a motorway passing through the Silesia region with over 50 active deep coal mines. The length of the Silesian part of the motorway is 45 km and the majority of it runs over the mining areas. The only material used for the embankment construction was anthropogenic soil. At this stage a decision was made that the impact of the mining influences on the motorway should be reduced and protective layers with geogrids should be designed [31]. Based on observed reduction in the pavement durability by 20% and more (higher designed deformations), it was necessary to apply solutions that minimize the impact of mining deformations. Due to a small range of the deformations, it was also necessary to mobilize the geosynthetics at the initial stage of the deformation.

Therefore, for the purposes of constructing the motorway in mining areas, a completely innovative approach was applied. The typical parameters for geogrids used for reinforcement are tensile strength $R_{max}$ [kN/m] and elongations at 2% and 5% ($R_{2\%}$ *i* $R_{5\%}$), respectively. Commonly the elongation at the breaking point $\varepsilon_{max}$ is also determined.

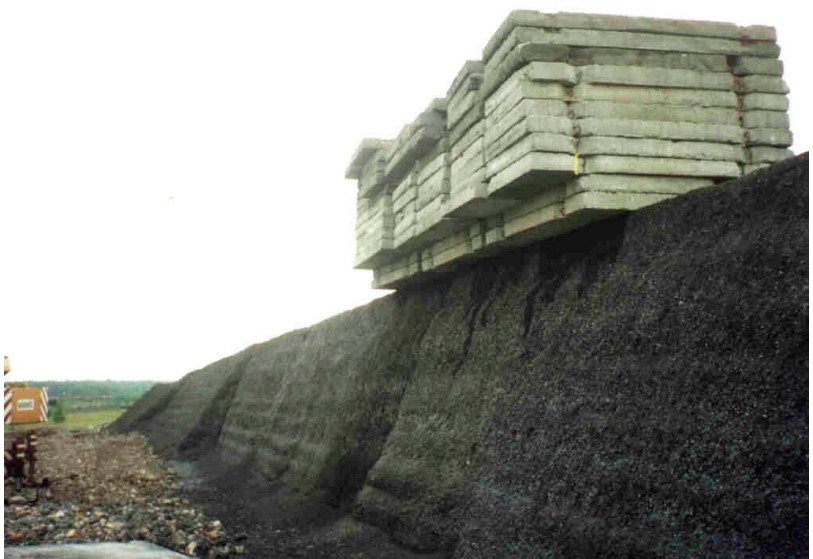

**Figure 6.** Full scale trial test to determine mechanical parameters of waste material.

In Figure 7, two extremely different behavioral models of geogrids are presented. Model (1) is characterized by high initial stiffness $E_i$ (1) and moderate strength $R_{max}$ (1). In contrast, the high strength $R_{max}$ (2) of model (2) is accompanied by low initial stiffness $E_i$ (2). Figure 7, shows a typical presentation of material characteristics, that usually are insufficient for a fully covered description of the material's mechanical performance. From the point of view of a proper protection of the motorway, mobilization at a very small deformation is very much of interest. Thus, the initial rigidity is $E_i$ tangential to the "force-elongation" curve, at the beginning of the stress–strain relationship, or the $E_S$ secant stiffness is a limited, deformable interval, up to 0.5%. The geogrid's built-in aggregate layer behaves differently to a geogrid tested in an in-situ condition.Thanks to mechanical compaction, the grains of sizes no bigger than the apertures get clogged in the apertures, practically eliminating the ability to deform. The stiffness of a geogrid built into the aggregate is; therefore, greater than the standard sample in the laboratory.

*2.3. Stabilization of Motorway Pavement*

Another example of a pioneer application of geomattresses is presented for the pavement stabilization in areas of deep coal mining. The construction of the pavement is as follows:

- 5 cm asphalt wearing course Stone Mastic Asphalt (SMA) 0/12.8 mm
- 10 cm asphalt base layer z Asphalt Concrete (AC) 0/25
- 12 cm asphalt sub base layer 0/31.5 mm
- 22 cm aggregate sub base layer from crushed unbound aggregate 0/31.5 mm,
- monolithic geogrid
- 20 cm technological layer made from natural unbound aggregate 0/31.5 mm
- 30 cm frost protective layer California Bearing Ratio (CBR) = 25%

The geomatress with two layers of geogrids installed under the embankment, used in the study, was the main solution to protect the motorway against the mining deformations. However, it was believed that the geomatress would not fully stop the deformation propagation, so another layer of

stabilizing geogrid was needed, filled within the pavement structure. The entire installation process is presented in Figure 8. A falling weight deflectometer (FWD) was used for further evaluation of the deformation process and effectiveness of the applied solution. The method is used to imitate actual conditions of traffic load. The FWD device measures the vertical deflection of the road pavement structure caused by falling of a fixed mass load from a set height. The mass falls naturally and transfers impulse to a bearing device. The vertical deflection induced by the plate is registered by a group of geophones placed in different distances from the axis. The load impulse value ranges from 7 to even 300 kN, depending on the type of pavement structure.

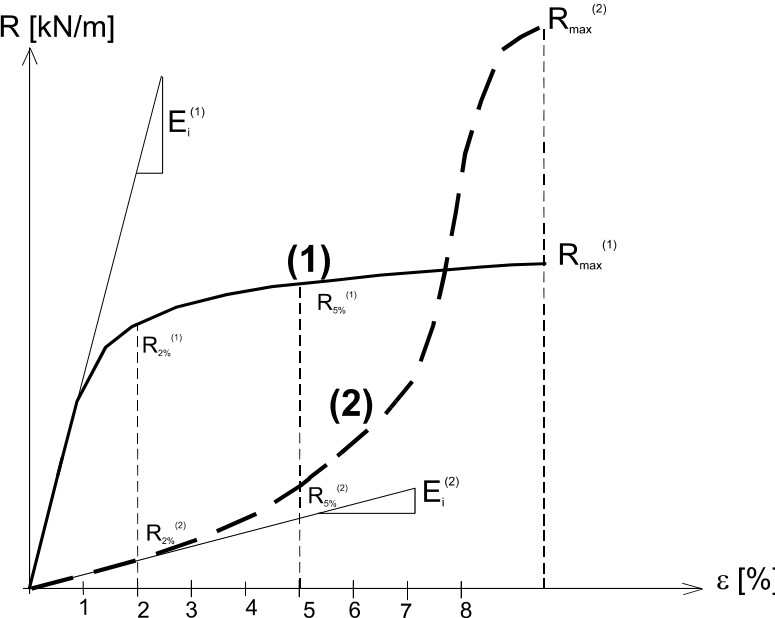

**Figure 7.** Stress–strain relationship for different types of geogrids: (1) high initial stiffness; (2) low initial stiffness.

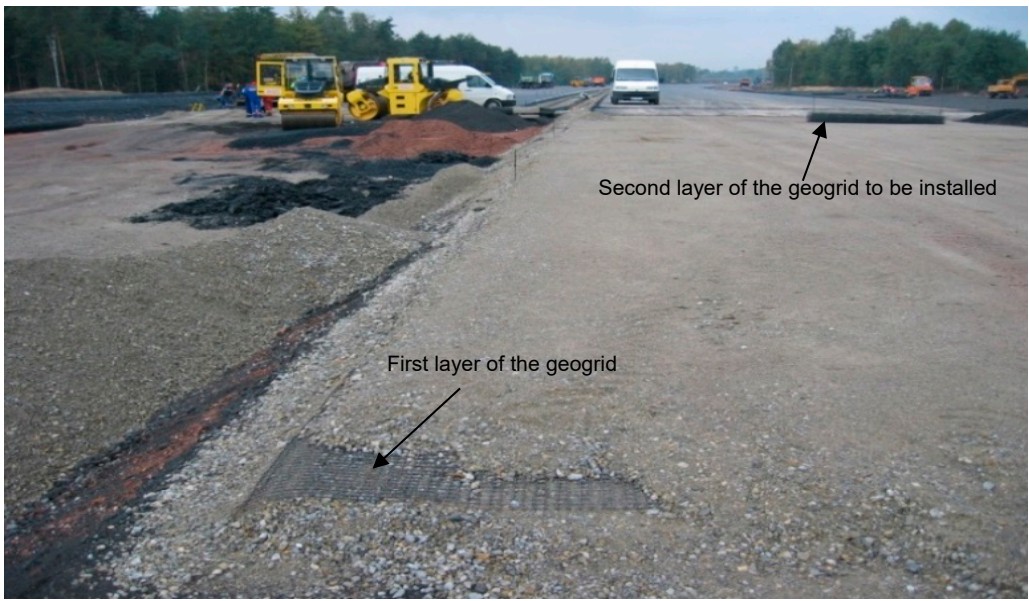

**Figure 8.** View of the pavement section where the geogrid was installed.

### 3. Results and Discussion on the Research Outputs

To understand how the geomattress could limit the influences from the mining activity, there was a sophisticated monitoring system designed on the pavement. As shown on Figure 9, two inclinometric-type pipes were installed beneath and above the geomattress. The site monitoring readings, collected simultaneously, showed significant differences in profiles beneath the geomatress, while the readings from the profile above were even. In some of the motorway sections, after the base was constructed, a significant ground subsidence took place. The biggest subsidence reached up 100 cm; however, the base profile above the geomattress was not seriously affected, which proved the protective role of the material used. A similar observation on the differences between deflection of deeper profiles were reported by Huckert et al. [35]. After the experiment on the first section, similar geomattresses were designed and installed for all other sections of the motorway, especially where the risk of mining subsidence was high.

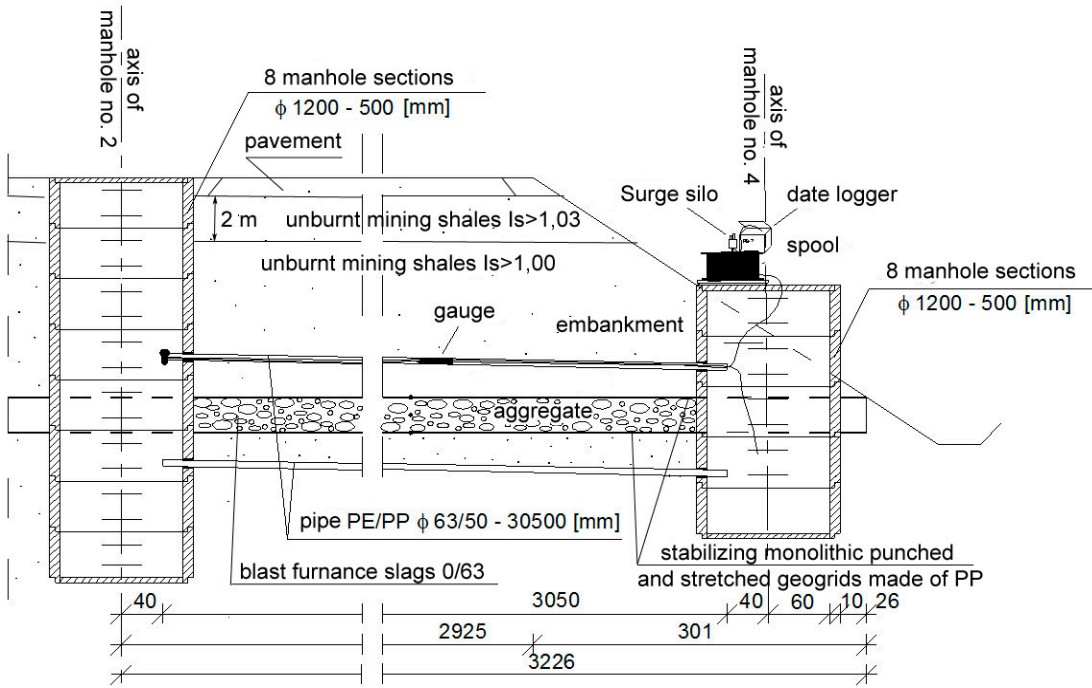

**Figure 9.** Cross-section of testing set designed to monitor geomattress deformations below the embankment.

Between years 1995 and 2008 the motorway sections in Silesia were completed, and until today the geomattresses helped keep the pavement in an undamaged condition. To provide a better understanding of collected deformation readings, a graph presenting the monitoring results for the first year of observation is given on Figure 10.

Bearing in mind that the permissible value of deformation is 1.5 mm/m, according to the standards of [7,20], the readings proved the sufficient bearing capacity of the used material. The mining works located near the control sections were carried out to the south of the motorway and they reached three different levels: level 418—wall 130 in years 2006–2007; level 502—wall 796 in year 2008; and level 504—wall 006 in years 2010–2011. The location of wall 006 and the measure subsidence is presented in Figure 11.

Taking into consideration the location of the excavation walls in the south, it is important to highlight that the motorway was subjected to continuous horizontal deformations in the west direction. When analyzing measured values of the subsidence, it was noted that the summarized vertical deformation of the north line reached 30 cm. The technical assessment indicated that the surface remained preserved in sufficient conditions, and only in two locations single longitudinal cracks were

observed. After 10 years of use, the surface was evaluated on the basis of the deflection measurements using FWD, The measurement was made in the right wheel track, only on the outer, on the most loaded lines.

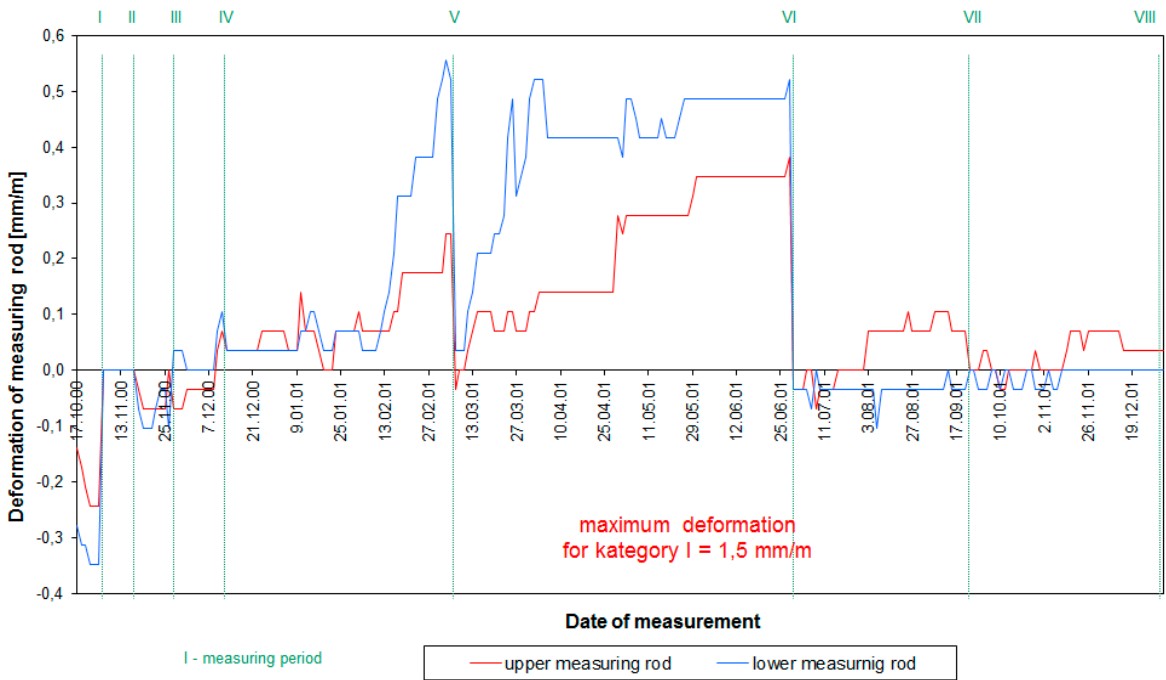

**Figure 10.** Analysis of strain changes in lower and upper (rods) ties of the geomattress built into the embankment, for the first year of exploitation.

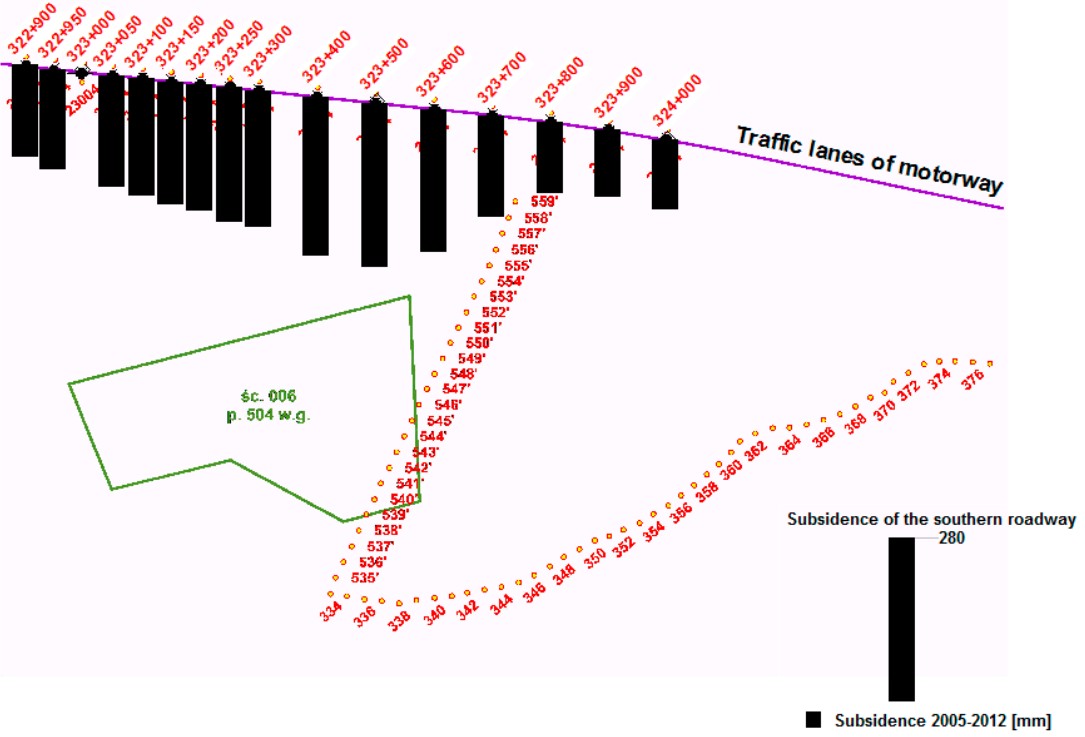

**Figure 11.** Subsidence of the southern roadway with the location of the 006 wall (300 m from the motorway).

Based on deflections measured every 50 m, the standardized deflections were compared with the permissible values for the surface with the highest traffic load (KR7) [27]. The results are presented in

Figure 12. The deflections were calculated based on corrected measured deflections of used testing force 50 kN, at temperature of 20 °C, taking into account the month of the measurement. The formula is given below:

$$DST = D_0 \cdot (50/F) \cdot f_T \cdot f_S \cdot f_P$$

where:

DST—standardized deflection,
$D_0$—measured deflection (FWD),
$f_T$—coefficient of temperature,
$f_S$—coefficient of season (September, $f_S$ = 1.20),
$f_S$—coefficient of subbase (flexible pavement, $f_S$ = 1.0).

The assessment of the pavement deflections (Figure 12) indicates the good technical condition of the pavement. Despite the 10-year period of use and additional load, caused by mining deformations, the calculated deflections were lower than the permissible values, assumed for the highest class of the technical condition. However, it should be noted that in the area from 323 + 800 km to 323 + 400 km, larger deflections were observed, especially on the southern section, which is closer to the so-called zone of the main impacts of mining operations. Such observations could also be found in researches conducted worldwide [36–38]. The major difference between the different studies is that the mining influenced area studied in the present paper was much more complex and specific for this particular region. Despite the difficult working conditions, the material allowed for the significant limitation of the subsidence effect.

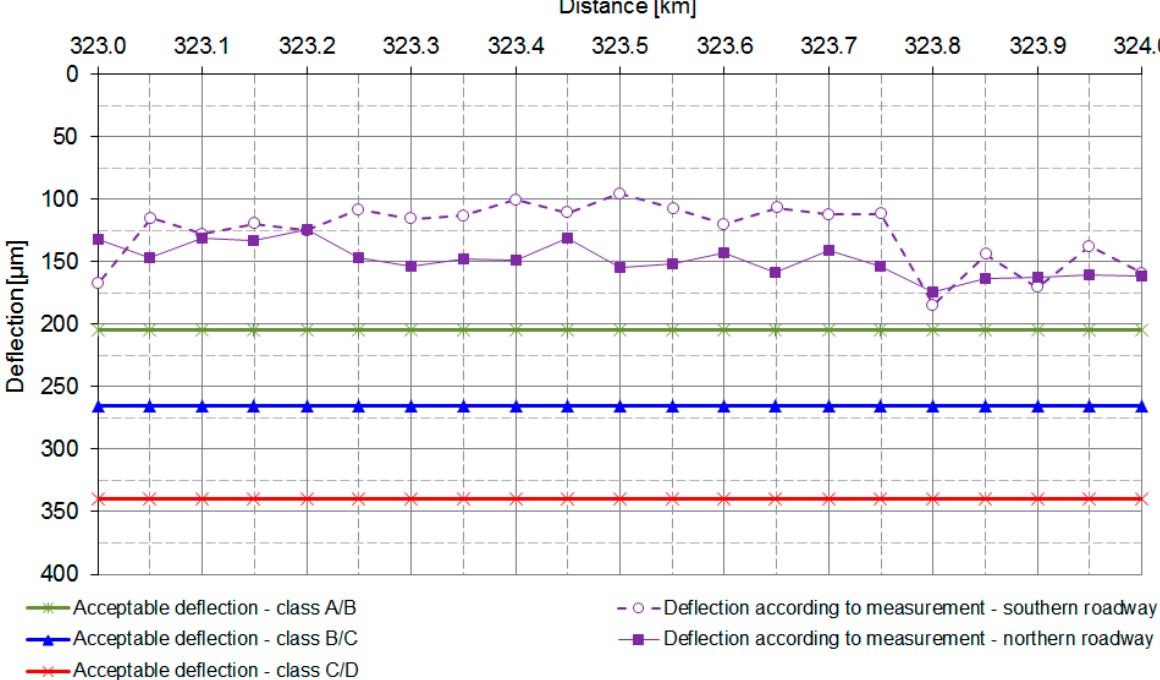

**Figure 12.** Measured (using Falling Weight Deflectometer) and permissible deflections of the motorway pavement loaded with KR7 traffic, after 10 years of exploitation.

Further analysis of the technical condition of the pavement was based on FWD results, by performing inverse calculations to determine the elastic modules for surface layers, using such parameters as loading magnitude, circular loading plain radius, and deformations measured using a geophone located within the loading axis. The results of calculations for each layer is presented in Figure 13.

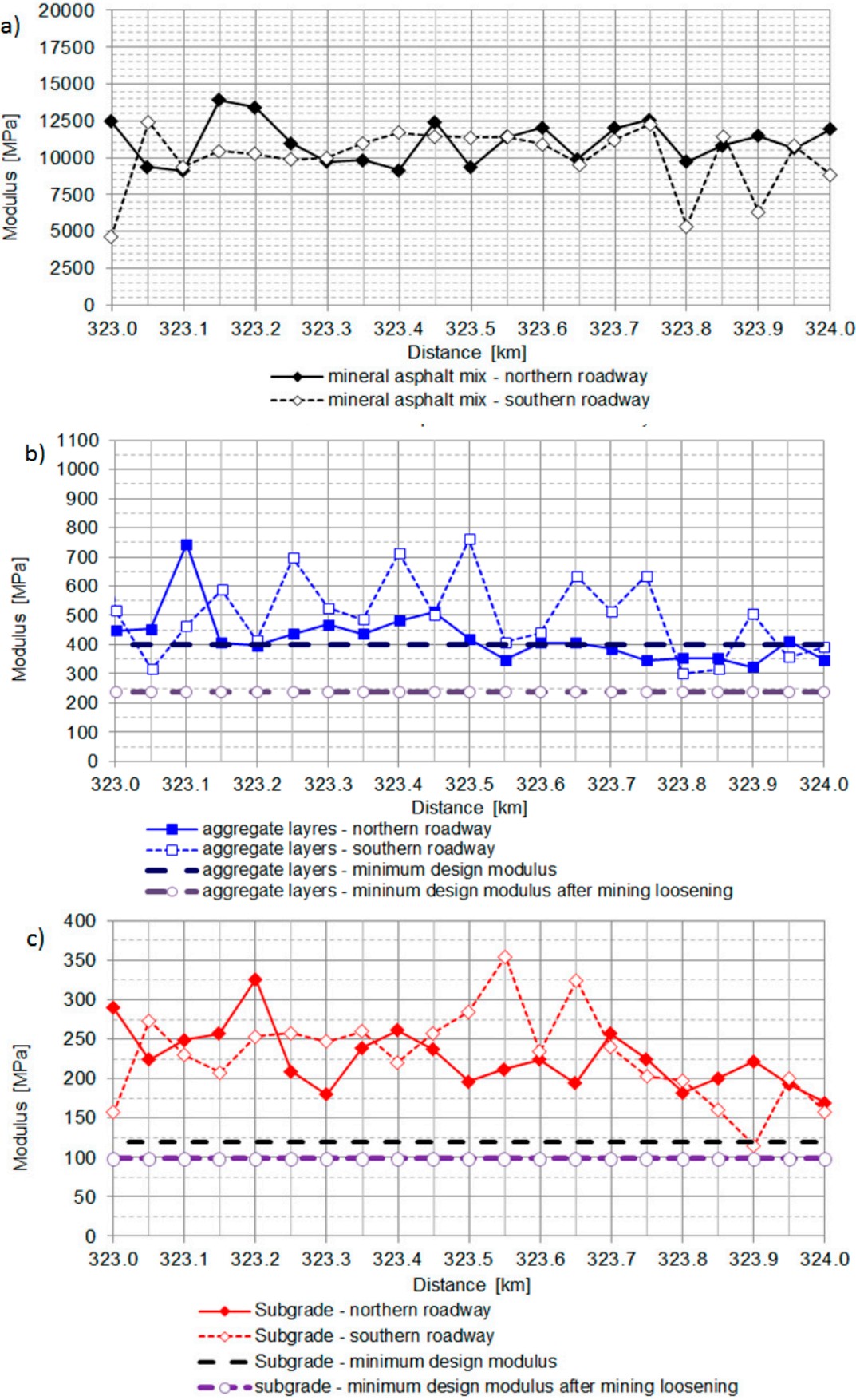

**Figure 13.** Calculation of moduli for: (**a**) Mineral asphalt mix, (**b**) aggregate layers, and (**c**) subgrade.

Detailed analysis of the values for the calculated pavement elastic modules indicated that the pavement and the subsoil were of high rigidity, which is in agreement with the assumptions made by Zornberg et al. [39] and Sudarsanan, et al. [40]. At the same time, in the section of increased deflections (i.e., from 323 + 800 km to 324 + 00 km—southern section), the value of modules was reduced. Whereas, the minimum modules did not exceed the minimum value, taking into account the loosening of the substrate with the mining deformations (Figure 13c).

The embankment sections, additionally, had a layer of aggregate stabilized by two layers of monolithic geogrids, in the form of geomattresses. The same solution was also included within the dipper layers of the pavement structure. The durability of the applied solution was proved by the values obtained from calculations on the basis of FWD field measurements [18]. To estimate the reduction of the module due to the loosening effect, a value of 1.5 mm/m was used. This value is in line with Polish Mining Area Category I.

## 4. Conclusions

In the present paper the geosynthetic applications in road construction in Poland were presented. The use of geosynthetics in the present study concerns the areas of mining influences. The paper aimed at proving the application reliability of such materials in linear infrastructures based on the presented case study.

Based on the laboratory tests, the conclusion was that a careful and reasonable determination of vertical deformation of the filing materials is crucial, especially for such working conditions as those met at post-mining sites. When it comes to the present case study, after over 15 years of use the solution still remains effective. The main conclusions drawn from the research were:

- The applied measurement system installed at the site allowed a qualitative assessment, indicating the reduction of deformations of the area where geosynthetic layers were applied.
- The assessment of the condition of the motorway pavement using the FWD and the calculated modules, based on the back analysis, confirmed the effectiveness of the geomattresses used. For the motorway pavement, a reduction of a maximum of 30 cm was revealed. Nevertheless, the aggregate layer module of the surface was not reduced. The confirmation of this conclusion was the lack of damage on the pavement itself.
- The results of the research confirm that mining exploitation with simultaneous use of the aggregate stabilization method with geogrid did not cause significant changes in the stiffness of the pavement layers. It seems that also in the area of the higher mining category, this conclusion can be considered appropriate. However, further research and monitoring needs to be performed at the site.

The application of geogrids in combination with anthropogenic materials has proved to be very efficient in terms of a deformation measurement. An important advantage is that tens of thousands of cubic meters of anthropogenic soils have been applied and used for the construction. Such a solution meets the European standards regarding environmental and sustainable development. All the observations, both field and laboratory tests, did not show a negative impact of anthropogenic soils on the properties and behavior of the geogrids used in the study.

**Author Contributions:** Conceptualization, E.K. and J.K.; methodology, J.K., M.G. and E.K., software, J.K., M.G., E.K. and P.O.; validation, J.K., M.G. and P.O.; formal analysis, J.K. and E.K.; investigation, M.G. and P.O.; resources, J.K. and E.K.; data curation, J.K., M.G. and P.O.; writing—original draft preparation, J.K., M.G. and P.O.; writing—review and editing, P.O.; visualization, J.K. and P.O.; supervision, J.K. and E.K.; funding acquisition, J.K. and E.K.

**Funding:** This research received no external funding.

**Conflicts of Interest:** The authors declare no conflicts of interest.

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
