# Peer review of "Lessons Learned on Geosynthetics Applications in Road Structures in Silesia Mining Region in Poland"

_applsci, doi:10.3390/app9061122_

Round 1

Reviewer 1 Report

The manuscript is interesting and informative, but there is a lack of data on the physical and mechanical properties of used the materials. Also the tested model should be drawing more detail in the figure 8.

The manuscript has some formatting inaccuracies ( spaces between numbers and measures units (100cm), does not use subscripts (130 row ....) and more.

Author Response

The authors would like to express their appreciation for all valuable suggestions and comments to improve the scientific quality of the manuscript.  The authors would like to assure that the paper was proof read, carefully reviewed, checked and corrected. Below you will find the specific answers to all given comments and suggestions.

Reviewer 1 comments:

The manuscript is interesting and informative, but there is a lack of data on the physical and mechanical properties of used the materials. Also the tested model should be drawing more detail in the figure 8.

Answer: The geotechnical and physical parameters for used materials are now provided in Tab. 1, and Fig. 8 (now  it is Fig. 9) is described in more details. Please see the corrected copy.

The manuscript has some formatting inaccuracies ( spaces between numbers and measures units (2500px), does not use subscripts (130 row ....) and more..

Answer: All the errors have been corrected.

Reviewer 2 Report

This manuscript being re-submission of a previously submitted paper, a lot of improvement was expected from the authors however, state of the paper is very much similar to the first submission. Very poor English throughout the manuscript, draft should be edited by a native English speaker. It is not clear if Figure 2, 3 are results from the case study presented in the draft. Order of the information provided in the Methodology is inappropriate. In the images provided, I could not work out where the mine is in respect to the pavement location. Throughout the paper, it was implied by the authors that numerical modelling was conducted but I could not find any numerical modelling results. The manuscript is like a consultancy report and minimal discussions were provided. None of the data presented from the report was discussed with data from previous studies. This work can be considered as a consultancy or educational report but I am afraid this is not a scientific article. The authors should ask themselves, what they have added to the existing science in this field, and I cannot find any.

Author Response

The authors would like to express their appreciation for all valuable suggestions and comments to improve the scientific quality of the manuscript.  The authors would like to assure that the paper was proof read, carefully reviewed, checked and corrected. Below you will find the specific answers to all given comments and suggestions.

Reviewer 2

This manuscript being re-submission of a previously submitted paper, a lot of improvement was expected from the authors however, state of the paper is very much similar to the first submission. 

Answer: The manuscript was restructured, the entire section on landfill case study was removed, so it is now focused on road construction case study only. The introduction has been completed by references and description on similar studies preformed in the past. Also the discussion was extended to meet the scientific publication standards. Please see the corrected copy.

Very poor English throughout the manuscript, draft should be edited by a native English speaker.

Answer: The manuscript has been reviewed and corrected by English native speaker. The authors believe that the present stile is acceptable for publication.

It is not clear if Figure 2, 3 are results from the case study presented in the draft. 

Answer: This has been explained in the main text. Figure 2 presents a general view of the subsidence causing factors, and it was prepared by authors based on their experience. Figure 3 presents the deformation ranges based on datasets collected from different case studies analysed by authors in the past case studies. The reference here has been added.

Order of the information provided in the Methodology is inappropriate.

Answer: The authors agree that trial loading test should go after the section on stabilization of the road embankment. This has been corrected and sections were switched. The Methodology section was restructured according the suggestion. We hope that was the issue the reviewer kindly pointed out.

In the images provided, I could not work out where the mine is in respect to the pavement location.

Answer: These is now explained in details in the main text. Most of the road embankment distance was located and overcame the mining exploitation grounds. Major area of Silesia region is post mining land, heavily exploited in the past.

Throughout the paper, it was implied by the authors that numerical modelling was conducted but I could not find any numerical modelling results.

Answer: The modeling part included only trial loading test, reflecting the in situ conditions. The crucial part was to obtained the mechanical properties of waste (given in the text). To avoid any confusion the use of models for such purpose was explained in the main text and the figure 8 has been added. It presents the FEM mesh, the geometry of the embankment with boundary conditions used for further evaluation. Please see the corrected copy.

The manuscript is like a consultancy report and minimal discussions were provided. None of the data presented from the report was discussed with data from previous studies. 

Answer: To improve the scientific soundness of the manuscript the discussion part has been extended and references for similar studies were provided. Please see the corrected copy of the manuscript..

This work can be considered as a consultancy or educational report but I am afraid this is not a scientific article. The authors should ask themselves, what they have added to the existing science in this field, and I cannot find any.

Answer: The new version of the manuscript has been corrected according the suggestions, thus the authors believe that the present copy meet the standards of a scientific publication. However, the authors are also fully aware of very much applicable (case study) character of the manuscript.

Reviewer 3 Report

The revised version is qualified for publication. 

Author Response

The authors would like to express their appreciation for all valuable suggestions and comments during the entire revision process.

Reviewer 4 Report

I have checked the paper again and I think the authors were able to restructure the paper and make it better. Therefore, I am for the publication of this ppaer.

Author Response

The authors would like to express their appreciation for all valuable suggestions and comments during the entire revision process. The authors would like to assure  that the paper was again proof read, carefully reviewed, checked and corrected.

Round 2

Reviewer 1 Report

The manuscript is interesting and it's quality is good.

Author Response

The authors would like to express their appreciation for valuable suggestions and comments throughout the entire revision process.

Reviewer 2 Report

Particular comment only on the results and discussions:

 Line 221-223: This is methods neither results nor discussions.

Line 228: There is a reference to profile above, I cannot see any dataset presented using a profile above this section.

Line 230-231: I do not understand this sentence.

Line 246-247: The authors speak about long-term evaluations of geomaterials however, one year worth of data is presented which cannot be related to long term quality analysis.

Line 252-25: I cannot relate the locations (e.g. level 418-wall 130) to a certain area of interest the authors are suggesting. A ma p should be provided supporting all these information.

Line261: There is no indication of FWD whatsoever in the methodology.

Line 272: Deflections are not calculated, they are measured, I believe.

Line 287: You are referring to (from 323 + 800km to 324 + 00km - southern section), where is this located? I cannot relate to this in your paper.

Line 298: There are massive information in Figure 12 but are not discussed in the manuscript.

Line 315: I cannot see any back-analysis

Line 324: Using geogrid to control deformation has been proved for many years, what is your research going to add to this field of science?

General comment:

English is poor, it MUST be proof read by a native speaker.

I am struggling very hard to find any novelty in this work.

The work is of absolute local importance, there is no international importance to it.

Author Response

The authors would like to express their appreciation for valuable suggestions and comments to improve the scientific quality of the manuscript. The authors would like to assure all the Reviewers that the paper was proof read, carefully reviewed, checked and corrected. Below you will find the specific answers to given comments and suggestions.

Reviewer 2

Particular comment only on the results and discussions:

Line 221-223: This is methods neither results nor discussions.

Answer: This is a general observation, indicating the pavement’s technical condition. The statement is provided to give a reader clear information on results of applied solutions. More discussion is given in following lines (l. 227-231).

Line 228: There is a reference to profile above, I cannot see any dataset presented using a profile above this section.

Answer: To avoid any misunderstanding the sentence was rewritten. The authors, by saying “profile above” meant the profile above the geomattress. It refers to Figure 10, where the differences of deformations between upper and lower rods for measurement periods IV-VI are significant. Please refer to corrected copy of the manuscript. It also needs to be highlighted the monitoring system of the study area was established in zones where the subsidence reached up to 100 cm, however these are not the values recorded directly above or beneath the geomattresses, but for the mean profile.

For such high values for the entire zone significant differences of deformations were observed along the geomattresses profiles (Fig. 10). The profile beneath the geomattress was much more deformed (blue line in fig. 10) than the profile below (red line).

Line 230-231: I do not understand this sentence.

Answer: As describe in previous answer. Significant subsidence became a measurement background for the profile deformation records.

Line 246-247: The authors speak about long-term evaluations of geomaterials however, one year worth of data is presented which cannot be related to long term quality analysis.

Answer: The results given in fig. 10 are those obtained in the first year of pavement exploitation. The next figures show the results recorded after 10 years, what was indicated in the text. Based on pavement’s deflection measurements (fig. 11) and calculated moduli analysis (fig. 12) its good technical condition is confirmed. The titles of figures were changed and more details were added in the main text.

Line 252-25: I cannot relate the locations (e.g. level 418-wall 130) to a certain area of interest the authors are suggesting. A ma p should be provided supporting all these information.

Answer: In the original version in the manuscript the map was provide, however to keep the editing and layout balance, it was decided to remove it from the paper. The present figure (fig. 11) gives a location of wall 006 in p. 504, for which the subsidence measurements have been collected for period 2005-2012. The wall was exploited in 2010-2011 and was located closely (300 m) to the motorway.

Line 261: There is no indication of FWD whatsoever in the methodology.

Answer: As recommended, the methodology section was completed by description of FWD method used for determination pavement’s deflections. The results of FWD test were presented in figure 11. To make it clearer to readers the title of the figure was also changed.

Line 272: Deflections are not calculated, they are measured, I believe.

Answer: Deflections are calculated. Fig. 11 presents standardized deflections (corrected measured deflections of used testing force 50 kN at temperature of 20°C, taking into account the month of the measurement). The formula for the calculation is now included in the main text. Please see the corrected copy of the manuscript.

Line 287: You are referring to (from 323 + 800km to 324 + 00km - southern section), where is this located? I cannot relate to this in your paper.

Answer: These are the distances of the analysed sections, they are presented on fig. 11 and then in fig. 12, after 10 years of pavement exploitation.

Line 298: There are massive information in Figure 12 but are not discussed in the manuscript.

Answer: In l. 287-289 it was stated that the reduction of moduli was observed. The overall reduction of moduli was lower than in case of the areas that were not reinforced using geosynthetics.

Line 315: I cannot see any back-analysis

Answer: In the paper the authors presented only the results of back analyses. The computations were performed using ELMOD software. It was not the authors intention, and it is not the purpose of the paper to present and describe the entire procedure required when performing the back analyses.

Line 324: Using geogrid to control deformation has been proved for many years, what is your research going to add to this field of science?

Answer: The authors would like to assure the reviewer that to the best of our knowledge there are no publications available showing the influence of such extensive mining influence (1 m settlemnts) on particular motorway construction layers. Also very much limited information on such long term (10 years) studies is available, based on comprehensive literature review conducted by the authors. The main reason for this is that the stabilisation function of geosythetics was adopted by ISO standard very recently, in October 2018.

General comment:

English is poor, it MUST be proof read by a native speaker.

Answer: Two external native speakers, familiar with the road construction topic, reviewed the manuscript. The authors believe that the present form is improved enough to be accepted for publishing.

I am struggling very hard to find any novelty in this work.

Answer: The novelty here is the investigation and performance assessment of the motorway for very high deformations caused by mining activities. The subsoil conditions in the present study are very much specific, as the coal mining for this area have been conducted for very shallow depths, thus the soil mechanical processes are very difficult to predict. For this reason 10 years monitoring of the site was performed to verify and confirm the effectiveness of the applied solutions.

The work is of absolute local importance, there is no international importance to it.

Answer: The mining activities are still being performed worldwide (Australia, China, Russia, Czech Republic, USA). The material presented in the manuscript is a meaningful study, describing prevention measures, that should be taken into consideration for such challenging tasks as road construction, on post mining areas.